# Mechanisms of RCD-1 pore formation and membrane bending

Keli Ren [1], James Daniel Farrell[1,2,3], Yueyue Li[4,5], Xinrui Guo[1,6], Ruipei Xie[1], Xin Liu[6], Qiaozhen Kang[6], Qihui Fan [1], Fangfu Ye[1,7], Jingjin Ding [4,5] & Fang Jiao [1] ✉

Regulator of cell death-1 (RCD-1) governs the heteroallelic expression of RCD-1-1 and RCD-1-2, a pair of fungal gasdermin (GSDM)-like proteins, which prevent cytoplasmic mixing during allorecognition and safeguard against mycoparasitism, genome exploitation, and deleterious cytoplasmic elements (e.g., senescence plasmids) by effecting a form of cytolytic cell death. However, the underlying mechanisms by which RCD-1 acts on the cell membrane remain elusive. Here, we demonstrate that RCD-1 binds acidic lipid membranes, forms pores, and induces membrane bending. Using atomic force microscopy (AFM) and AlphaFold, we show that RCD-1-1 and RCD-1-2 form heterodimers that further self-assemble into ~14.5 nm-wide transmembrane pores (~10 heterodimers). Moreover, through AFM force spectroscopy and micropipette aspiration, we reveal that RCD-1 proteins bend membranes with low bending moduli. This combined action of pore formation and membrane deformation may constitute a conserved mechanism within the broader GSDM family.

Gasdermins (GSDMs) and gasdermin-like proteins are a family of pore-forming proteins responsible for pyroptosis, an inflammatory type of programmed cell death, and have been widely identified and characterized in mammals[1-7], early vertebrates[8], invertebrate animals[9], and bacteria[10,11]. A typical GSDM comprises a cytotoxic N-terminal domain, which exposes intrinsic membrane pore-forming activity, and an inhibitory C-terminal domain (in animals) or fragment (in bacteria), which acts as an N-terminal repressor[8,9,12-16]. Upon activation by proteolytic cleavage of the two components (typically, but not exclusively, by an inflammatory caspase), the N-terminal domains assemble into cyclic oligomers which form pores in the cell membrane, thereby effecting cell death.

GSDMs are abundant and widespread in fungi[17-20], with some species encoding more than 20 GSDM homologs in their genomes[18]. While some fungal GSDMs follow the same route of proteolytic activation as their animal and bacterial cousins, *regulator of cell death-1*

(RCD-1) in the filamentous fungus *Neurospora crassa* (*N. crassa*) is unusual. RCD-1 is regulated by the co-expression of alleles from the two haplogroups RCD-1-1 and RCD-1-2, which encode sequences of 257 and 244 amino acids, respectively[17,18]; cell death only occurs when *both* proteins are present. Structurally, they strongly resemble the N-terminal domains of human and murine GSDMs, and both lack an inhibitory C-terminal domain. The mechanism of RCD-1 activation and subsequent pore formation are not yet known.

The pore formation mechanisms and propagates of GSDMA3, GSDMD and GSDMB have been investigated by various techniques including cryo-electron microscopy (cryo-EM)[4-7], atomic force microscopy (AFM)[21-23], and molecular dynamics simulations (MD)[24,25]. Complexes formed by these proteins share a conserved pore structure comprising anti-parallel β-barrels which are compactly aligned to form the transmembrane region[4-6,26,27]. Monomers first assemble into arc-shaped oligomers on the membrane surface which then, in some

[1]Beijing National Laboratory for Condensed Matter Physics, Institute of Physics, Chinese Academy of Sciences, 100190 Beijing, PR China. [2]School of Physical Sciences, University of Chinese Academy of Sciences, 100049 Beijing, PR China. [3]Songshan Lake Materials Laboratory, Dongguan, Guangdong 523808, PR China. [4]National Laboratory of Biomacromolecules, CAS Center for Excellence in Biomacromolecules, Institute of Biophysics, Chinese Academy of Sciences, 100101 Beijing, PR China. [5]University of Chinese Academy of Sciences, 101408 Beijing, PR China. [6]School of Life Sciences, Zhengzhou University, Zhengzhou 450001, PR China. [7]Oujiang Laboratory (Zhejiang Lab for Regenerative Medicine, Vision and Brain Health), Wenzhou Institute, University of Chinese Academy of Sciences, Wenzhou, Zhejiang 325000, PR China. ✉e-mail: fang.jiao@iphy.ac.cn

order, (1) grow into slit-shaped or ring-shaped oligomers and (2) insert their β-hairpins into the membrane[21–25].

Here, we apply AFM to investigate the activation and oligomerization of RCD-1. To further understand the interactions of RCD-1 and lipids, we characterize RCD-1 binding, insertion, and oligomerization with 54 different lipid compositions. By using AFM with high spatial resolution and Alphafold, we illustrate that RCD-1-1 and RCD-1-2 co-activate by forming a heterodimer. Using time-lapse AFM, we reveal the mechanisms of RCD-1 assembly and pore formation. Our images show that RCD-1 assembles into arc- and slit-shaped oligomers, which form dynamic structures that transform into larger, thermodynamically stable ring-shaped oligomers. Finally, we observe that RCD-1 induces lipid membrane deformation beyond pore formation, which is controlled by RCD-1 density and membrane bending modulus.

## Results

### RCD-1-1/RCD-1-2 heterodimer is the basic unit of RCD-1 oligomers

In mammalian GSMDs, pore formation requires the assembly of activated N-terminal domains into ring- or slit-shaped oligomers, whose parallel β-hairpins insert into the membrane. Since (1) no proteolytic activation step has been observed in RCD-1 and (2) no pore formation occurs when only RCD-1-1 or RCD-1-2 is present, we hypothesize that RCD-1 pore complexes comprise both RCD-1-1 and RCD-1-2 monomers.

We used AlphaFold2-multimer and AlphaFold3 to predict the structures of RCD-1-1 homodimer, RCD-1-2 homodimer, and RCD-1-1/RCD-1-2 heterodimer (Fig. S1, S2). Both homodimers are predicted to assemble into symmetric complexes with non-parallel β-hairpins. On the other hand, the β-hairpins in the asymmetric heterodimer complex arrange in a parallel fashion, producing a structure more like a mammalian GSDM dimer and better disposed to simultaneous membrane insertion.

To test these predictions, we incubated RCD-1-1, RCD-1-2, and mixture of them on mica for 5 min (Fig. 1a(i)(ii)(iii)). AFM imaging revealed numerous small oligomers on the surfaces in all instances, indicating that both homo- and heterocomplexes can form. Notably, while larger oligomers form in a mixture of the two proteins (Fig. 1a(iii)), there are no arcs, slits, or rings.

Experiments with model membranes reveal a different picture. We prepared supported lipid bilayers (SLBs) of *E. coli* total lipid on a freshly cleaved mica surface in a buffer solution (see "Materials and Methods"). For all the SLBs used in this work, we confirmed by AFM imaging that the bilayers were defect-free and covered the mica surface uniformly (Fig. S3b). We then incubated the SLB with 250 nM RCD-1 (RCD-1-1 and/or RCD-1-2) for 5 min at room temperature (Fig. 1a(iv-vi)). After rinsing with buffer solution, we imaged the SLBs with AFM in either tapping mode or peak-force tapping mode. AFM images of SLBs containing only one of RCD-1-1 or RCD-1-2 showed no signs of protein aggregation, suggesting that individual proteins of either type lack the capacity to interact with the membrane, or the interaction too weak to anchor proteins in a fixed position on the membrane during scanning (Fig. 1a(iv)(v)).

By contrast, 1:1 mixtures of RCD-1-1 and RCD-1-2 formed arc-, slit- and ring-shaped oligomers (Fig. 1a(vi)). Oligomers in the pore and pre-pore states were indistinguishable by height and mobility (Figs. 1b, c). Following previous reports on GSDMs[4,5,22], we categorize RCD-1 pre-pores as including both membrane-attached mobile oligomers, and membrane-inserted oligomers in the plugged pre-pore state, in which the inner membrane has not yet been released. RCD-1 pores consist of membrane-inserted oligomers that form open transmembrane pores with inner membrane released. Negative-stained transmission electron microscopy (TEM) results corroborate the pore structures (Fig. S4a). The number of oligomers increased in proportion with concentration (Fig. S4b). Ring oligomers dominate the population (91.6 ± 1.7% of oligomers) with only small numbers of arcs (3.2 ± 1.0%) and slits

(5.2 ± 1.3%, Fig. 1d) observed. The upper protrusion (top ring) diameters of pre-pore and pore structures vary widely from 5.2 to 27.7 nm, measuring 14.5 ± 3.2 nm on average (n = 1582), smaller than those observed in GSDMD (23.1 ± 0.7 nm)[21] or GSDMA3 (26.3 ± 2.2 nm)[22].

We detect both immobile (n = 1282) and mobile (n = 300) rings (Fig. 1e). We infer that mobile rings are adsorbed to but not inserted into the membrane, much like the pre-pore state reported for GSDMD and GSDMA3[4,5,22]. While mobile rings exhibit similar heights to immobile rings (2.9 ± 0.3 nm for mobile rings *vs.* 3.1 ± 0.3 nm for immobile rings; Fig. 1e(i)), their diameters are significantly smaller (10.1 ± 1.6 nm *vs.* 15.6 ± 2.5 nm; Fig. 1e(ii)). This difference may be due to differing numbers of monomers, or to different monomer spacings in an oligomer; further structural characterisations with higher spatial resolution could address this question.

To investigate the compositions of the oligomers, we conducted further experiments with various concentration ratios of RCD-1-1 and RCD-1-2, with individual concentrations ranging from 250 nM to 1 µM (Fig. 1f(i)-(v)). In these experiments, the lower concentration component was consistently maintained at 250 nM. Structures of oligomers in arc-, slit- and ring-shape formed in all cases (Fig. 1f(i)-(v)). The surface coverage remained consistently around 8.6% across all tested ratios, with values of 8.6 ± 0.6%, 8.7 ± 0.7%, 8.6 ± 0.5%, 8.7 ± 0.6% and 8.6 ± 0.5% observed in conditions of Fig. 1f(i)-(v), respectively (Fig. 1f(vi)), indicating that the degree of oligomerization governed is by the component present at a lower concentration. The results suggest arc-, slit- and ring-shaped oligomers comprise a 1:1 ratio of the two proteins, and, based on the Alphafold predictions, formed from heterodimers.

We then used high spatial-resolution AFM to characterize the ring pores formed by RCD-1. We successfully determined subunit structures (Fig. 1g, Fig. S5). Using this information, we determined the number of component dimers in ring-shaped structures formed in the lipid membrane. The perimeter of ring-shaped oligomers ranges from 16.2 to 87.1 nm, with an average of 45.8 ± 9.9 nm (n = 1582), corresponding to the count of RCD-1 dimers varies from 3 to 17 with an average of 9 ± 2 (Fig. 1h). Among the ring-shaped oligomers, the perimeters of immobile rings range from 31.4 to 87.1 nm with average of 49.0 ± 7.7 nm, corresponding to 6 to 17 dimers with average of 10 ± 2 (Fig. 1h). In contrast, the perimeters of mobile rings range from 16.2 to 46.9 nm with average of 31.9 ± 5.2 nm, corresponding to 3 to 9 dimers with average of 6 ± 1 (Fig. 1h). Compared to immobile rings, mobile rings are approximately 4 dimers smaller, which suggests that they may represent an intermediate state of RCD-1 ring-shaped pores, which could undergo further assembly and transformation to form final, larger pore structures.

Additionally, during the submission of our paper, the RCD-1 pore structure was resolved (Fig. 1g, upper right, Fig. S5)[28], showing that the RCD-1 pore is composed of 11 RCD-1-1/RCD-1-2 heterodimers, which falls into the range we observe for immobile rings (10 ± 2 dimers). Our high spatial-resolution topography also detected rings consisting of 5 dimers (Fig. 1g, Fig. S5d). Although we could not confirm whether these structures represent a pre-pore or pore state, they are consistent with the AlphaFold3-predicted RCD-1 pore structures composed of 5 RCD-1-1/RCD-1-2 heterodimers (Fig. 1g, bottom right, Fig. S5). This suggests that RCD-1 may form very small pores comprising only 5 heterodimers.

Our investigations suggest that RCD-1-1/RCD-1-2 heterodimer is the basic unit of RCD-1 oligomers. However, due the formation of homo-oligomers on mica (Fig. 1a), we cannot rule out the possibility that homodimers play some role in RCD-1 pore formation.

### Real-time imaging of RCD-1 oligomerization and pore formation

Time-lapse AFM imaging of a 1: 1 mixture of RCD-1 proteins with concentration 125 nM, on an *E. coli* total extract SLB reveals two distinct pore assembly processes. In one pathway, mobile oligomers grow into larger arcs and slits, eventually forming mobile rings (Fig. 2a). This pre-

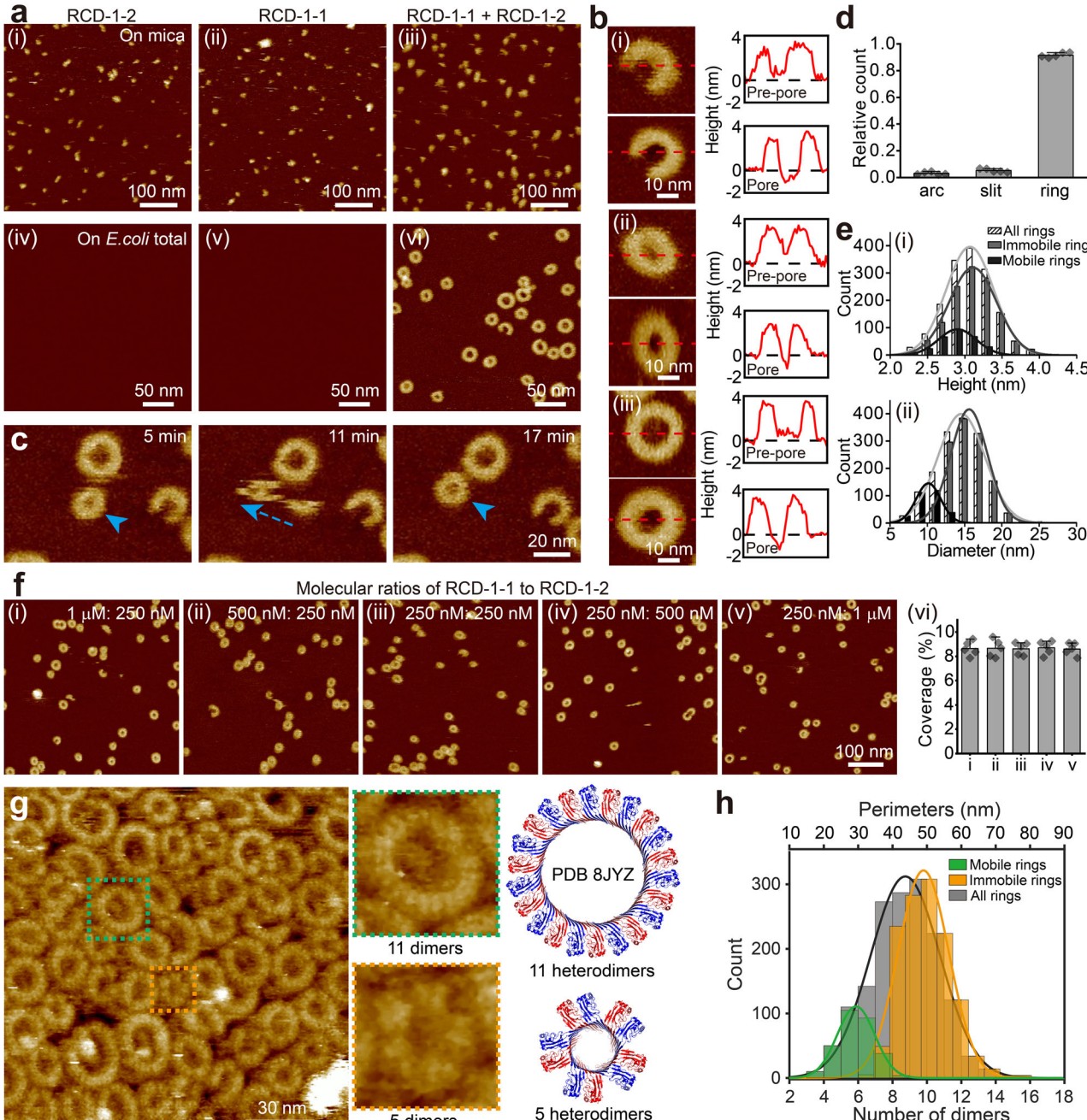

**Fig. 1 | RCD-1-1/RCD-1-2 heterodimers assemble into oligomers and pores.**
**a** AFM topographies showing oligomerization of RCD-1-1, RCD-1-2, and their mixtures on mica (i–iii) and *E. coli* total lipid membranes (iv–vi). Experimental repeats: i-v, *n* = 3; vi, *n* = 20. **b** Representative (i) arc-, (ii) slit-, and (iii) ring-shaped oligomers observed in prepore and pore states, with height profiles marked by red lines (experimental repeats, *n* = 20). Black dashed lines indicate the SLB surface. **c** Mobile RCD-1 rings visualized on lipid membranes, with blue arrows tracing representative trajectories (experimental repeats, *n* = 20). **d** Quantification of arc-, slit-, and ring-shaped oligomers across five independent experiments reveals a predominance of ring structures (Mean ± SEM). **e** Height (i) and diameter (ii) distributions of

immobile and mobile rings, fitted with Gaussian curves to determine the mean ± std value. **f** AFM imaging of RCD-1 oligomers on SLBs with varying RCD-1-1:RCD-1-2 ratios (i-v, 1 μM:250 nM to 250 nM:1 μM). (vi) Surface coverage remains consistent (-8.6%). Data are presented as mean values ± SEM from 5 independent experiments. **g** High-resolution AFM topography of RCD-1 oligomers on SLBs composed of DOPE: CL(6: 4). Highlighted oligomers (green/orange) contain 11 and 5 heterodimers, respectively, corresponding to cryo-EM and AlphaFold3-predicted pore structures. **h** Distribution of ring perimeters and dimer counts: immobile rings average 49.0 ± 7.7 nm (10 ± 2 dimers, mean ± SD), mobile rings average 31.9 ± 5.2 nm (6 ± 1 dimers).

pore state may persist for tens of minutes (Fig. 2b) before eventual membrane insertion and lipid expulsion. In the alternative assembly pathway, short oligomers of as few as two RCD-1 dimers insert into the membrane, thus becoming immobile (Fig. 2c). Upon growing to around six dimers in size, pore formation and lipid release occurs. These immobile assemblies may continue to oligomerize, resulting in the formation of ring-shaped pores (Fig. 2c).

## RCD-1 oligomers lead to membrane bending

Lipid compositions have been reported to play a critical role in GSDM oligomerization[21,22,24,29]. Here, we investigate RCD-1 oligomerization with various lipids compositions. No oligomer was observed when co-incubation of 250 nM RCD-1-1 and 250 nM RCD-1-2 on mica SLBs made by neutral lipids (pure DOPC or 1: 1 DOPC: DOPE) (Fig. S6). On SLBs comprising DOPC and DOPS at mass ratios 6: 4 and 8: 2, co-incubation

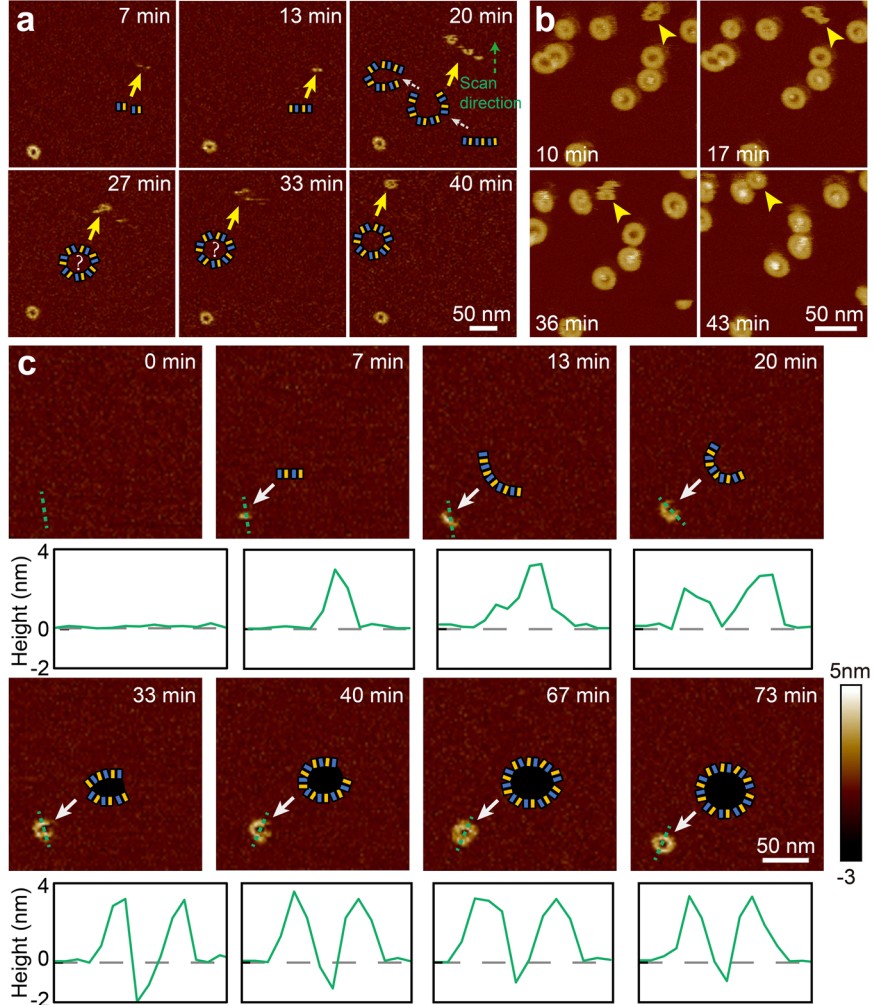

**Fig. 2 | Time-lapse topographs showing RCD-1 oligomerization and pore formation. a** Time lapse topographies showing the high mobility of RCD-1 oligomers during oligomerization, with data collected by incubation of 125 nM RCD-1-1 and 125 nM RCD-1-2 on *E. coli* total SLB (experimental repeats, *n* = 3). The cartoons underneath to indicate the interpretation of growing oligomers. The yellow arrows indicate the same oligomers, the green arrow represents the scanning direction of AFM imaging. **b** The mobile rings transform into pores, with data collected by incubation of 500 nM RCD-1-1 and 500 nM RCD-1-2 on *E. coli* total SLB. The yellow arrowheads indicate the same oligomers (experimental repeats, *n* = 3). **c** Time-lapse topographs showing immobile RCD-1 oligomerization and pore formation, with data collected by incubation of 125 nM RCD-1-1 and 125 nM RCD-1-2 on *E. coli* total SLB (experimental repeats, *n* = 3). The bottom panels show the height profile along the green dashed line. The yellow arrows indicate the same oligomers. The cartoons indicate the interpretation of growing pores.

of RCD-1-1 and RCD-1-2 not only results in the formation of arcs, slits, and ring-shaped oligomers, but also induces membrane bending—an effect not previously reported in studies of other GSDMs (Fig. 3a(i)(ii), Fig. 4, Fig. S7–S8). Bent membranes exhibit a variety of shapes, ranging from small, sphere-like structures to large, irregular, flat forms (Fig. 3a(i)(ii), Fig. 4, Fig. S7–S9). Whether a sphere-like or flat defect is observed depends to some extent on the magnitude of the imaging force; when using larger forces, the defects appear to be flatter (Fig. S7). To minimize parachuting artifacts that can produce long tails, we typically employ imaging forces ranging from 40 to 80 pN. The bent membranes are all some ~5 nm higher in elevation than the unperturbed membrane, and rougher, due to the accumulation of small particles on their surfaces (Fig. 3a). Ring-shaped oligomers tend to localize at the edges of the defect, with their ring planes making a non-zero angle with the mica substrate (Figs. 3a, 4c), indicating membrane bending. Membrane defects, bordered by multiple end-to-end connected RCD-1 arcs, were observed to localize within the bent membrane (Fig. S8b, d).

RCD-1 pore formation and membrane deformation still occur upon substituting DOPS with other negatively charged lipids (we tried DOPG, cardiolipin (CL), phosphatidylinositol (PI), and phosphatidic acid (PA); Fig. S10-16). DOPG, in compositions DOPC: DOPE: DOPG = 6: 2: 2 or 5: 3: 2, leads to comparable results to those with lipids containing PS (Fig. 3a (iii), Fig. S10, S11). Additionally, numerous small particles were observed on the surface of the bent membrane (Fig. 3a (iii), Fig. S11). The sizes of these particles are approximately 3 nm in height and ~5.5 nm in width, indicative of RCD-1 dimers (Fig. 3a (iv) (v)). Thus, we speculate that membrane bending is induced by the interaction with RCD-1 dimers.

RCD-1 also induces bending in giant unilamellar vesicles (GUVs). Using confocal fluorescence microscopy, we imaged fluorescently labeled GUVs with composition DOPC: DOPS: [Liss rhod PE(18:1)] = (8: 2: 0.005) in a solution with 1 μM FITC-dextran. Within 4.3 s of exposure to 500 nM RCD-1, the spherical GUV had developed a cone-like protrusion with a locally decreased concentration of PE (Fig. 3b). This observation suggests that RCD-1 affects local membrane composition in addition to membrane curvature.

To estimate the bending moment exerted by RCD-1 on the GUV, we employed micropipette aspiration to measure the bending

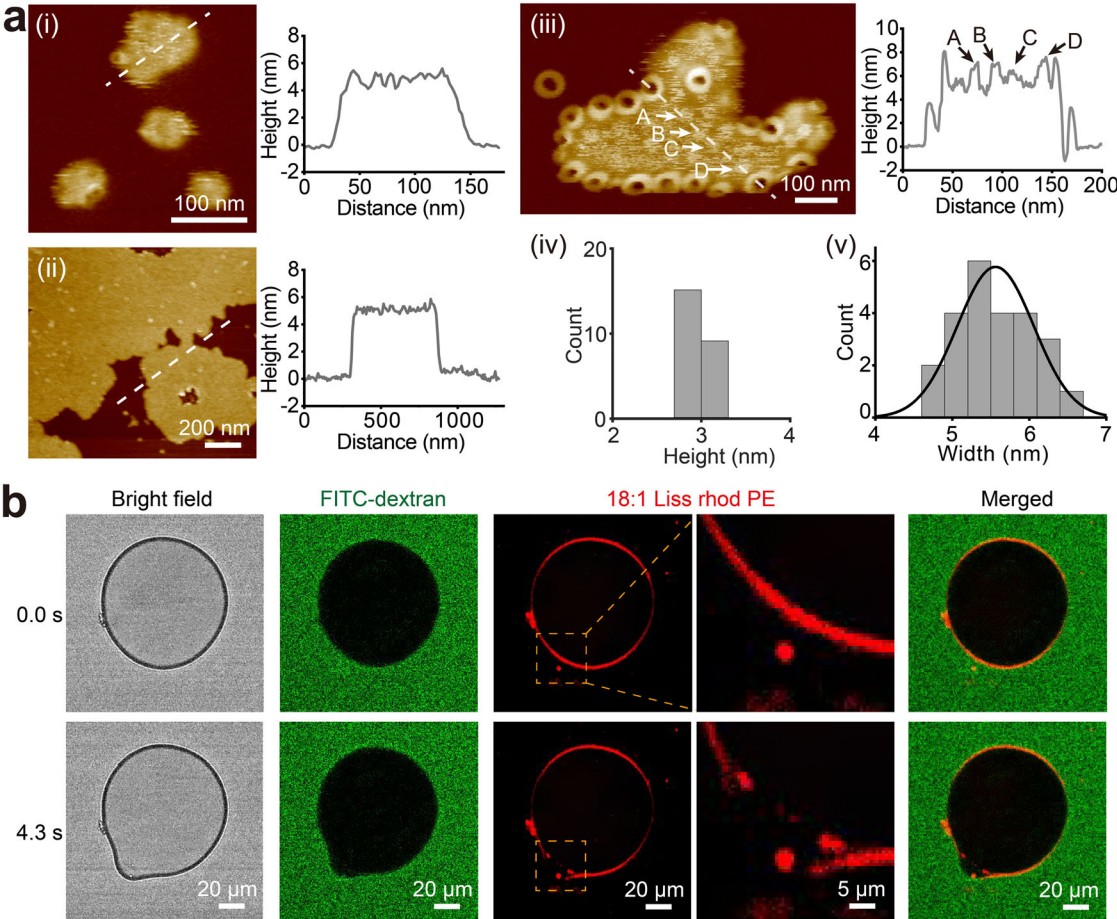

**Fig. 3 | Membrane bending induced by RCD-1. a** Bent membranes of DOPC: DOPS = 6: 4 or DOPC: DOPS = 8: 2 ranging from small, sphere-like structures (i) to large, irregular, and flat forms (ii) (i-ii, experimental repeats, $n$ = 10). (iii) High-resolution AFM image of the bent membrane of DOPC: DOPE: DOPG = 5: 3: 2, with small particles on the surface (A, B, C, D) (experimental repeats, $n$ = 3). The particles occur at heights of 3.0 nm ($n$ = 24) (iv) and are 5.6 ± 0.5 nm (mean ± SD, n = 24) in width (v). The right panels show the height profile along the dashed line. **b** Confocal fluorescence microscope images of GUV local membrane bending upon addition of RCD-1. The GUV composition is DOPC: DOPS: Liss rhod PE(18:1) = 8: 2: 0.005. Green represents FITC (dextran) in the solution and red represents 18:1 Liss rhod PE of GUV (experimental repeats, $n$ = 3).

modulus of the GUVs (Fig. S17). The curvature of a surface can be characterized by the average curvature ($1/R_1 + 1/R_2$), where $R_1$ and $R_2$ are the principal radii of curvature measured along two perpendicular directions[30]. For a spherical surface, $R_1 = R_2$. The bending moment $M$ of the curved membrane can be calculated using the formula:

$$M = k_c(1/R_1 + 1/R_2) \qquad (1)$$

where $k_c$ is the bending modulus. The method for calculating the bending modulus of GUVs follows previous studies[31–33]. The measurements yielded a bending modulus of 18.1 ± 1.2 $k_B$T for the GUVs formed by DOPC: DOPS = 8: 2, the same components used in Fig. 3b. The original radius of the GUV $R_G$ is approximately 46 μm, while the radius of the membrane bending region induced by RCD-1 $R_B$ is about 15 μm (Fig. 3b). The change in curvature during this process is ($1/R_B - 1/R_G$). Therefore, the bending moment exerted by RCD-1 on the GUV $M_{RCD-1}$ can be calculated by $M_{RCD-1} = k_c \times (1/R_B - 1/R_G)$[30], resulting in a value of approximately 0.8 $k_B$T/μm.

**In situ imaging of membrane bending**

Time-lapse AFM reveals that membrane bending begins at lipid defects stabilised by RCD-1 oligomers, whether in the form of arcs, rings, or multiple end-to-end connected oligomers (Fig. 4a). Upon further addition of 250 nM RCD-1 (RCD-1-1 and RCD-1-2) monomers, bent

membranes expand along the edges of defects, growing and merging with one another until they cover the entire screen area (Fig. 4a). Later removing the monomers causes membrane bending to cease, with some of the bent areas reverting to their original SLB state (Fig. 4b). Notably, RCD-1 oligomers on the bent membrane remained mobile, and new RCD-1 ring structures formed at the edges of the bent membranes (Fig. 4c). This suggests the RCD-1 oligomers that mediate membrane bending may remain in an intermediate oligomeric state, which could further assembly into more stable ring structures through diffusion to the edges of bent membranes and interaction with smooth SLBs.

A two-step membrane bending was observed upon introducing a low concentration of RCD-1 to the bent area. As depicted in Fig. 5, 50 nM RCD-1 were introduced to replace the pre-incubated 250 nM RCD-1 on lipid membrane of DOPC: DOPS = 8: 2 at 10 min. Initially, expansions of the bent area were observed with a height increase ~2 nm, followed by a subsequent height increase to a final height of 4 to 5 nm, accompanied by a reduction in the size of the bent area. We refer the initial 2 nm height increase as the intermediate state of membrane bending, corresponding to lower RCD-1 dimer density (Fig. 5a, yellow circles). The subsequent further height increase is attributed to the condensation of RCD-1 dimers. The results indicate RCD-1 concentration plays an important role in membrane bending.

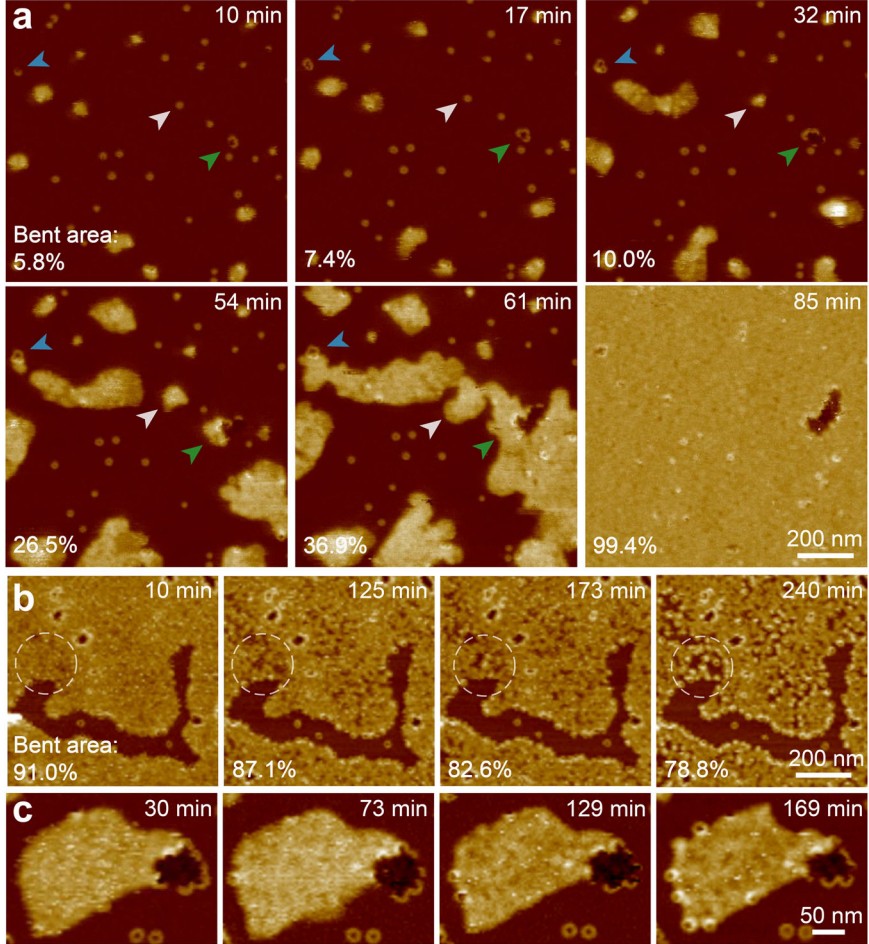

**Fig. 4 | AFM characteristics of membrane bending. a** Time lapse topographies showing the process of RCD-1-induced membrane bending, with data collected by incubation of 250 nM RCD-1-1 and 250 nM RCD-1-2 on DOPC: DOPS = 6: 4. The blue, white and green arrowheads indicate the membrane bending initiated from pores of arc, ring and end-to-end connected oligomers. The percentage of bent membrane areas shows 5.8%, 7.4%, 10.0%, 26.5%, 36.9%, and 99.4% at times of 10, 17, 32, 54, 61 and 85 min, respectively (experimental repeats, $n = 5$). **b** Time lapse topographies showing that some of the bent areas (partial highlighted by white circular pattern) reverting to their original SLB state upon removing the monomers at 10 min. The percentage of bent membrane areas shows 91.0%, 87.1%, 82.6%, and 78.8% at times of 10, 125, 173 and 240 min, respectively (experimental repeats, $n = 3$). **c** New RCD-1 ring structures formed at the edges of the bent membranes upon removing the monomers at 30 min (experimental repeats, $n = 5$).

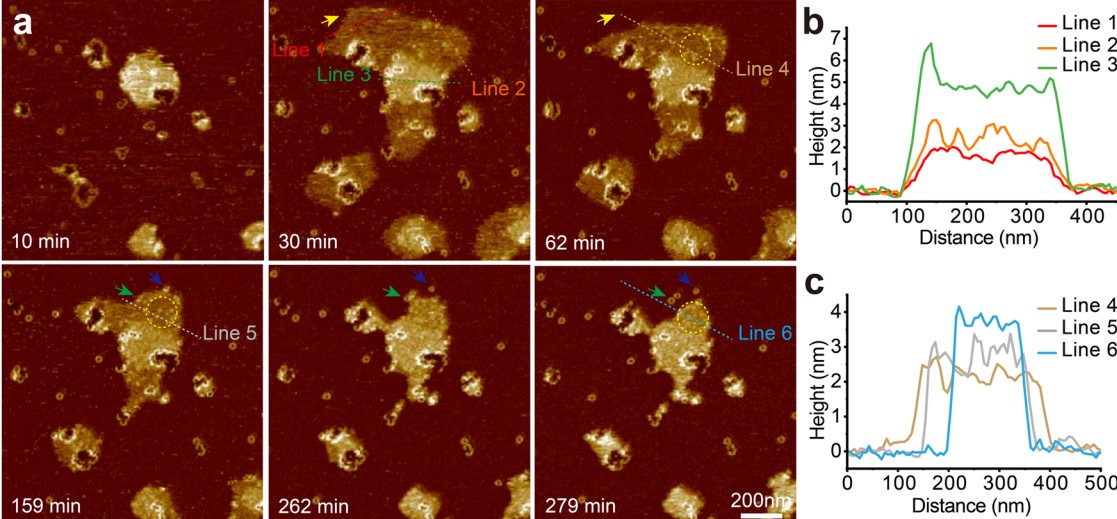

**Fig. 5 | Time-lapse AFM reveals a two-step membrane bending process at low concentration of RCD-1. a** Membrane bending underwent curvature expansion at first, followed by an increase in curvature accompanied by a reduction in the size of bent area (experimental repeats, $n = 5$). Specifically, 250 nM RCD-1 was co-incubated with DOPC: DOPS = 8: 2 to form some initial bending islands, followed by a switch to 50 nM RCD-1 at 10 min. Yellow arrows and circles represent the same position, while green and blue arrows represent new formed rings. **b**, **c** The height profiles of the indicated membrane bending sites.

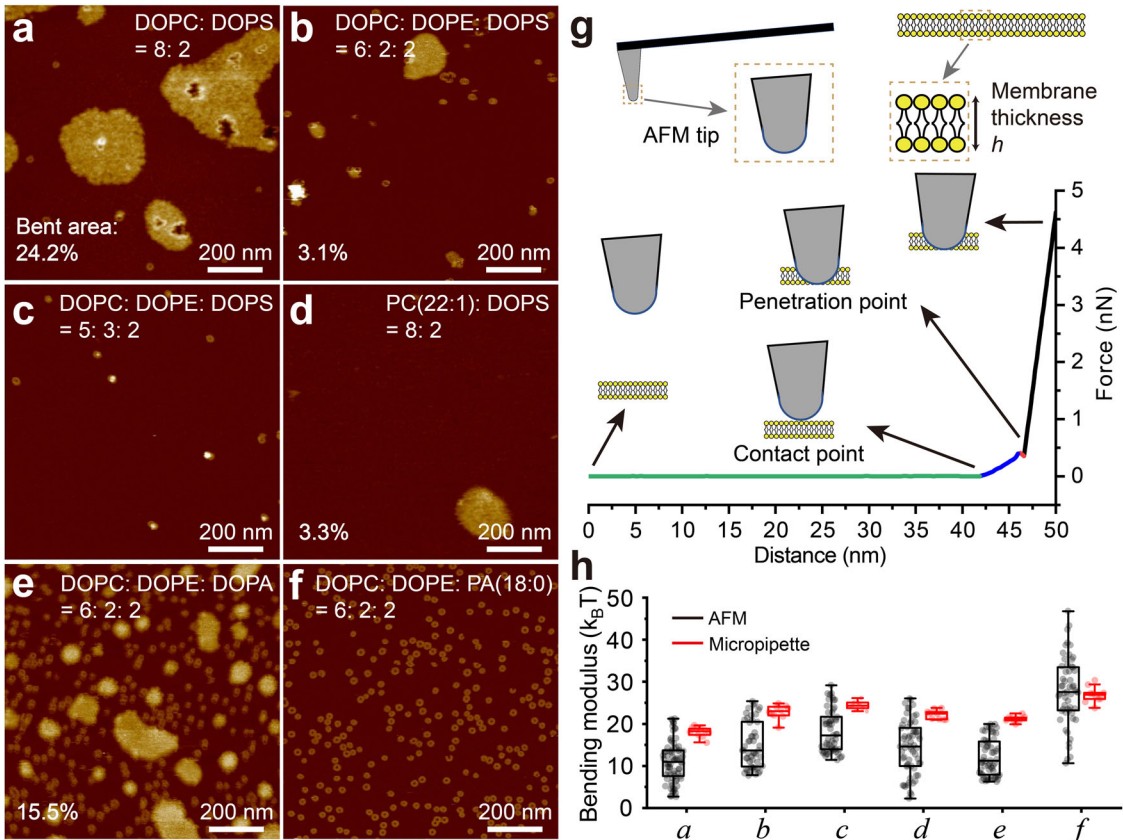

**Fig. 6 | Membrane stiffness measurement by AFM force spectroscopy and micropipette aspiration.** AFM topographs of RCD-1 incubated with various lipid membranes of DOPC: DOPS = 8: 2 (**a**) (**a**, experimental repeats, $n = 10$), DOPC: DOPE: DOPS = 6: 2: 2 (**b**), DOPC: DOPE: DOPS = 5: 3: 2 (**c**), PC(22:1): DOPS = 8: 2 (**d**), DOPC: DOPE: DOPA = 6: 2: 2 (**e**), DOPC: DOPE: PA(18:0) = 6: 2: 2 (**f**). **b–f** Experimental repeats, $n = 3$. **g** Illustration of how force spectroscopy was used to measure membrane rigidity. **h** Bending moduli of the six lipid compositions, measured by AFM force spectroscopy (black) and micropipette aspiration (red). Symbols $a$-$f$ represent the lipid components of (**a**–**f**). The AFM force spectroscopy experiments included $n = 73, 51, 55, 62, 59$, and $56$ force curves for $a$-$f$, respectively. For micropipette aspiration experiments, $n = 10, 10, 9, 9, 9$, and $10$ GUVs for $a$-$f$, respectively. In box plot visualizations, the central line represents the median, while the bottom and top edges of the box denote the 25th and 75th percentiles, respectively.

**Membrane rigidity plays a significant role in membrane bending**

The extent of bending is also sensitive to membrane composition. For example, after incubating 250 nM of RCD-1-1 and RCD-1-2 on a (DOPC: DOPE: DOPS) = (8: 0: 2) membrane for 5 min, ~24% of the membrane bends (Fig. 6a). Substituting a quarter of the DOPC with DOPE (6: 2: 2 DOPC: DOPE: DOPS) reduces the extent of bending by a factor of eight (Fig. 6b), while further increasing the weight fraction of DOPE (5: 3: 2 DOPC: DOPE: DOPS) eliminates bending altogether (Fig. 6c, Fig. S9). Ring-shaped pores still form in all cases. We see similar results for other acidic lipids: for weight fractions (DOPC: DOPE: DOPG) = (2: 6: 2), (DOPC: DOPE: Heart CL) = (2: 6: 2), and (DOPC: DOPE: DOPI) = (4: 4: 2), almost no bending is observed; reducing the (DOPE: DOPC) ratio recovers the bending phenomenon (Fig. S10, S12, S13, Supplementary Table 1). Since PE is known to increase membrane rigidity[34–36], we hypothesized that a membrane composed of stiffer lipids would also exhibit less bending; using the stiffer saturated PA(18:0) instead of unsaturated DOPA(18:1) in a (6: 2: 2) mixture of (DOPC: DOPE: PA) preserves pore formation but inhibits bending, suggesting that membrane rigidity does indeed determine the extent of membrane bending (Fig. 6e, f).

To quantify our hypothesis that membrane rigidity determines the degree of membrane bending, we measured the bending moduli of some of the SLBs used in this work with AFM force spectroscopy, using the AFM tip to press and penetrate the SLBs (Fig. 6g, "Materials and Methods")[37,38]. The results revealed that the bending modulus of DOPC: DOPS = 8: 2 was $11.1 \pm 4.9$ $k_BT$, whereas DOPC: DOPE: DOPS = 6: 2: 2 exhibited a higher modulus of $15.0 \pm 5.5$ $k_BT$ (Fig. 6h). Upon increasing the PE component to 30% with DOPC: DOPE: DOPS = 5: 3: 2, the bending modulus rose to $18.3 \pm 5.0$ $k_BT$ (Fig. 6h). Under this condition, pore structures were observed, while membrane bending was almost absent.

Interestingly, when incubating with a thicker lipid membrane of (PC (22:1): DOPS) = (8: 2), we see a decrease in both pore formation and membrane bending compared to DOPC: DOPS = (8: 2) (Fig. 6a, d). The bending modulus increased ~3 $k_BT$ to $14.4 \pm 6.3$ $k_BT$ compared with (DOPC: DOPS) = (8: 2), which can explain the decrease in membrane bending (Fig. 6h). The results also suggest that the pore formation of RCD-1 is related to membrane thickness in addition to membrane acidity.

We also measured the bending moduli of GUVs with identical lipid compositions using micropipette aspiration by applying suction pressure to them (Fig. 6h, Fig. S17, "Materials and Methods")[31–33]. The values obtained from AFM and micropipette aspiration are qualitatively similar, although the average bending moduli measured by AFM were generally lower than those from micropipette aspiration (Fig. 6h). One possible explanation is that the GUV measurements using micropipette aspiration were conducted in a solution of 80% 100 mM sucrose and 20% 100 mM glucose, whereas our AFM measurements were performed in a salt buffer (20 mM HEPES-NaOH, pH 7.4, 150 mM NaCl). Numerous studies have shown that the bending modulus of

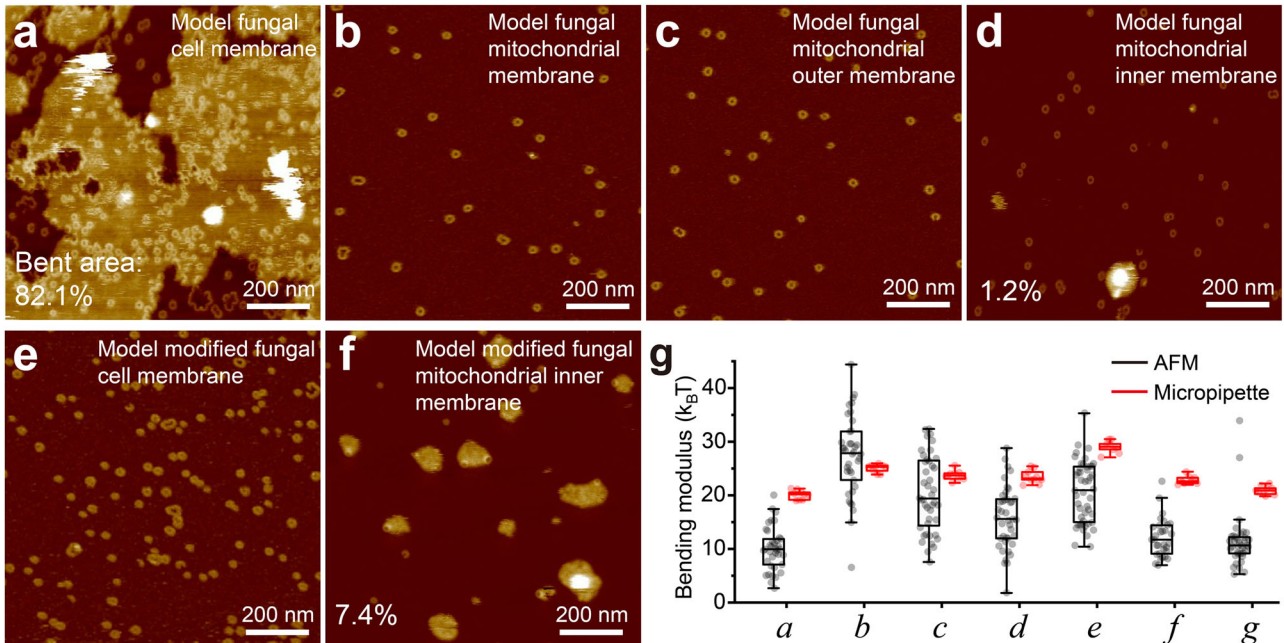

**Fig. 7 | RCD-1 oligomerization in fungal membranes.** AFM topographies of RCD-1 incubated with various different fungal membranes: model fungal cell membrane-DOPC: DOPE: DOPA: DOPS: CL(18:1) = 30: 28: 20: 4: 18 (**a**) (weight ratio); model fungal mitochondrial membrane-DOPC: DOPE: DOPA: DOPI: DOPS: CL(18:1): DOPG = 37: 43: 1: 8: 4: 6: 1(**b**); model fungal mitochondrial outer membrane-DOPC: DOPE: DOPA: DOPI: DOPS: CL(18:1): DOPG = 40: 36: 4: 9: 5: 5: 1(**c**); model fungal mitochondrial inner membrane-DOPC: DOPE: DOPA: DOPI: DOPS: CL(18:1): DOPG = 32: 30: 1: 5: 5: 25: 2(**d**); model modified fungal cell membrane- DOPC: DOPE: PA(18:0): DOPS: CL(18:1) = 30: 28: 20: 4: 18(**e**); model modified fungal mitochondrial inner membrane- DOPC: DOPE: DOPA: DOPI: DOPS: Heart CL(18:2): DOPG = 32: 30: 1: 5: 5: 25: 2(**f**) (**a**–**f**, experimental repeats, *n* = 3). Membrane bending occurred in the model fungal cell membrane (**a**) and model fungal mitochondrial inner membrane

(**d**). Change the PA from 18:1 to 18:0 for artificial fungal cell membrane, bending disappeared in model modified fungal cell membrane (**e**). Change the CL from 18:1 to 18:2 (Heart CL), more bending occurred in the model modified fungal mitochondrial inner membrane (**f**). **g** Bending moduli of the fungal membranes, measured by AFM force spectroscopy (black) and micropipette aspiration (red). Symbols *a-f* represent the lipid components of (**a**–**f**), while symbol *g* represents the yeast extract polar lipid. The AFM force spectroscopy experiments included *n* = 37, 38, 44, 41, 47, 31, and 37 force curves for *a-g*, respectively. For micropipette aspiration experiments, *n* = 9, 9, 10, 10, 9, 10, and 9 GUVs for *a-g*, respectively. In box plot visualizations, the central line represents the median, while the bottom and top edges of the box denote the 25th and 75th percentiles, respectively.

acidic lipid membranes decreases with increasing salt ion concentration[39,40]. Given that all lipid compositions we consider contain an acidic lipid, this effect may account for the difference in bending moduli measured by the two techniques. However, both AFM and micropipette aspiration measurements consistently demonstrate that membrane rigidity significantly impacts the degree of membrane bending.

### Pore-formation and membrane bending with fungal membranes

To gain deeper insights into the functions of RCD-1 in fungi, we modeled fungal cell membranes (DOPC: DOPE: DOPA: DOPS: CL(18:1) = 30: 28: 20: 4: 18) and mitochondrial membranes using previously reported components[41–59] (see "Materials and Methods"). Pore formation is evident in all our models (Fig. 7a–d), but we only see extensive bending in our model of a fungal cell membrane (Fig. 7a), suggesting that membrane bending has a particular biological relevance to pyroptotic-like cell death in fungal cells. As before, increasing the saturation of PA by substituting PA(18:0) for PA(18:1), leads to less bending in the cell membrane model (Fig. 7a, e). Some bending occurs on the mitochondrial inner membrane (Fig. 7d), substituting doubly unsaturated for monounsaturated CL in this model predictably increases the extent of bending (Fig. 7f). Both force-spectroscopy and micropipette aspiration measurements indicate that the model fungal cell membrane (Fig. 7a) and the model modified fungal mitochondrial inner membrane (Fig. 7f) exhibit lower bending moduli compared to the other model membranes (Fig. 7g). These findings support the conclusion that RCD-1-induced membrane bending occurs specifically in soft membranes.

Additionally, we investigated RCD-1 with a real fungal cell membrane: the yeast extract polar membrane. As shown in Fig. S18, membrane bending also occurred when RCD-1 was co-incubated with this membrane. AFM and micropipette aspiration measurements indicate that the yeast extract polar membrane has a bending modulus close to the values observed for the model fungal cell membrane and the model modified fungal mitochondrial inner membrane (Fig. 7g). This result aligns with our findings from artificial lipid membranes, reinforcing the robustness of our conclusions.

## Discussion

Daskalov et al. found that co-incubation with RCD-1 proteins leads to the disintegration of acidic liposomes[18]. Most importantly, RCD-1 also controls a rapid lytic cell death in *N. crassa*, which was the main reason and phenotype used to identify the gene[17,18]. Based on this behavior, and a strong homology to the cytotoxic N-terminal domains of human and murine GSDMs, the authors identified RCD-1-1 and RCD-1-2 as gasdermin-like proteins[18]. However, they found no direct evidence of oligomerization to form membrane pores, despite finding that the proteins form aggregates.

Here, we find that, on SLBs containing acidic lipids (PS, PG, CL, PI or PA), RCD-1 does form pore structures like those formed by other GSDMs, and via a similar mechanism: either by growing an oligomer from a dimer already inserted in the membrane, or by forming an oligomer which adsorbs to the membrane and subsequently inserts (Fig. 8). Furthermore, we observe that pore formation on acidic SLBs only occurs when both proteins are present. Our Alphafold calculations predict that only the heterodimer adopts a conformation

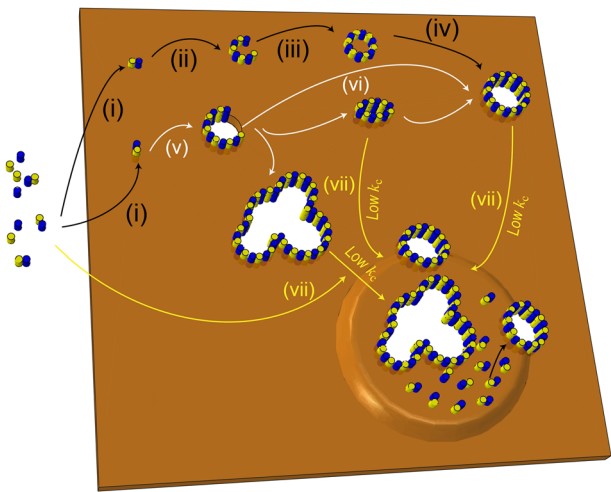

**Fig. 8 | Model of RCD-1 pore formation and bending membrane.** (i) RCD-1-1 and RCD-1-2 coactivated to form heterodimers. (ii-iv) These dimers oligomerize into mobile arc-, slit- and ring-shaped mobile structures before penetration into the membrane to initiate pore formation. (v-vi) The dimers oligomerize into immobile arc-, slit- and ring-shaped oligomers which have penetrated the membrane and continue assembling until a stable pore is formed. (vii) Membrane bending occurs when dimers bind to a membrane with low bending modulus, particularly in areas with membrane defects or pores.

homologous to GSDM dimers, which, along with our extensive stoichiometric studies, suggests the heterodimer as the smallest building block of RCD-1 pores.

While we do not observe membrane disintegration, we do find that, unlike other GSDMs, incubation with RCD-1 leads to small membrane bubbles and detachment of the membrane from the mica substrate, observed as an increase in membrane elevation ~ 5 nm. We find stiffer membranes are less affected by this distortion, with no distortion occurring in membranes with a bending modulus above ~18.3 $k_B$T (Fig. 8), implying that RCD-1-induced membrane bending has a role in the formation of both kinds of defect. In addition to the bending modulus, the curvature of membrane bending is also related to the number of proteins bound per unit area. When the bending modulus is constant, the greater the number of proteins bound per unit area, the more pronounced the membrane bending (Fig. 5). These factors may be responsible for the liposome disintegration observed by Daskalov et al.; in particular, the RCD-1 concentration used in that work (~3 μM) is more than ten times that used here. Extensive bending is also observed on membranes designed to mimic fungal cell membranes and on natural yeast extract polar membranes, suggesting that the bending phenomenon may have some biological function. We suggest that in fungal pyroptosis, RCD-1 not only forms pores in the cell membrane but may also act to pull the membrane away from the cell wall, undermining the structural integrity of the cell and causing cell death. In reality, both processes will depend on the concentration of RCD-1 proteins in the cell, as well as the specific interactions between the cell membrane and cell wall.

In summary, we not only observed the structure of the oligomers formed by RCD-1 in SLBs, but also demonstrated that these oligomers primarily consist of heterodimers of RCD-1-1 and RCD-1-2. In addition, we observed membrane bending phenomena that were not previously reported in studies of gasdermin proteins with SLBs. We have found that the bending modulus of SLBs plays a crucial role in regulating the interaction between RCD-1 protein and lipid bilayers. This implies that we can potentially modulate the impact of RCD-1 protein on membrane behavior by manipulating the composition of cell membranes. Although it remains unclear whether other GSDMs exhibit similar

effects on lipid bilayers, these findings provide a potential strategy for modulating cellular pyroptosis.

## Methods

### Protein expression and purification
The recombinant RCD-1-1 and RCD-1-2 proteins were separately expressed in *E. coli* BL21 (DE3). After lysis of the bacteria, supernatants were subjected to Ni²⁺-affinity chromatography. RCD-1-1 proteins were then purified by HiTrap SP cation-exchange and then Superdex G75 gel-filtration chromatography, while the RCD-1-2 proteins were purified by HiTrap Q anion-exchange and subsequently Superdex G75 gel-filtration chromatography. Proteins were stored at -80 °C in 20 mM Tris-HCl, 150 mM NaCl, pH 8.

### Preparation of supported lipid bilayers (SLBs)
The method for SLBs formation was detailed in our previous studies and followed a standardized procedure[60–62]. In brief, the lipids obtained from Avanti Polar Lipids were dissolved in chloroform, mixed with designed components. All the lipid compositions used in this work showed with mass ratio. The mixture was then dried using argon flow in a small glass vial, followed by >2 h incubation in a vacuum desiccator. Subsequently, lipids were fully rehydrated with buffer (20 mM HEPES-NaOH, pH 7.4, 150 mM NaCl) for 5 min at room temperature, with the volume adjusted to achieve a lipid solution at 0.5 mg/ml. Next, the lipid suspension was either bath-sonicated for 45 min or tip sonicated for ~4 min to obtain clear liposome solution. Finally, 10 μl of lipid solution was deposited onto a freshly cleaved 2.2 mm diameter mica disc, incubated for over 60 min to form fully covered SLBs and rinsed thoroughly with buffer (20 mM HEPES-NaOH, pH 7.4, 150 mM NaCl). Before incubation with RCD-1, SLBs were checked by AFM to ensure that they were defect-free and completely covered the surface. Only the SLBs with no defects were applied.

### AFM image
Except for the specific protein concentration and incubation time mentioned, ex-situ samples were prepared by incubating 250 nM RCD-1 (RCD-1-1, RCD-1-2, or a mixture of both) on SLBs or mica for 5 min, followed by thoroughly rinsing with 20 mM HEPES-NaOH, pH 7.4, 150 mM NaCl. Time-lapse AFM images of Fig. 2 were recorded using a mixture of 125 nM RCD-1-1 and 125 nM RCD-1-2 on *E. coli* total lipids. Time-lapse AFM images of Fig. 4 were recorded using a mixture of 250 nM RCD-1-1 and 250 nM RCD-1-2 on SLBs of DOPC: DOPS = 6: 4. AFM images (512 × 512 pixels) were acquired using a Nanoscope 8 (Bruker) in tapping mode or Peakforce QNM mode in buffer of 20 mM HEPES-NaOH, pH 7.4, 150 mM NaCl with tips of either SNL-10-B (Spring constant: 0.12 N/m, oscillation frequency: 2.5 ~ 3.5 kHz) or Scanasyst-Fluid+ (Spring constant: 0.7 N/m, peakforce frequency: 2 kHz) at room temperature. Scanning force was kept ≤80 pN to prevent sample perturbation. The scanning rate was set 1 ~ 2 lines/s. Date was analysed using Nanoscope software. High spatial resolution images were collected at 512 × 512 pixels and a scan speed of 2 Hz over a 150 nm range. Linear flattening and 3 × 3 median filtering were applied to reduce noise.

### Negative-stain electron microscopy
For negative-stain TEM, a mixture of 5 μl 10 μM RCD-1-1 and 5 μl 10 μM RCD-1-2 were incubated with 0.5 μl 1 mg/ml *E. coli* total liposomes at room temperature for 30 min. Subsequently, 1.5 μl of the mixture was dropped onto a glow-discharged copper grid coated with parlodion and carbon and left for 10 min at room temperature. The grid was then washed with 10 droplets of distilled water, followed by staining with 7 μl 1 % uranyl acetate for 10 s and blotting dry with filter paper. Negative-stained images were collected using JEOL2100 microscope with 200 kV.

## AFM force-spectroscopy

Membrane rigidity was measured by force spectroscopy with tip of Scanasyst-Fluid+ (spring constant: 0.7 N/m). To avoid damaging of the tip, the calibration procedures were performed at the end of force-spectroscopy experiment. To reduce the influence of the tip shape and substrate on the force curves acquisition and calculation, only the data from the region where the indentation depth does not exceed 1 nm were used for analysis and computation. For each measurement, over 50 positions with 10 curves per position were measured.

## Calculate the bending modulus $k_c$

The bending modulus $k_c$ of a film can be calculated by the formula as it is reported:

$$k_c = \frac{Eh^3}{12(1 - v^2)} \tag{2}$$

where $E$ is the Young's modulus, $h$ is the thickness of the membrane, and $v$ is the Poisson's ratio[37,38].

And the Young's modulus can be calculated by the Hertz model form force-indentation curve with a correction of bottom-effect artifact proposed by Garcia[63]:

$$F = \frac{4}{3} \frac{E}{1 - v^2} R^{1/2} \delta^{3/2} \left[ 1 + \frac{1.133\sqrt{\delta R}}{h} + \frac{1.497\delta R}{h^2} + \frac{1.469\delta R\sqrt{\delta R}}{h^3} \right.$$
$$\left. + \frac{0.755(\delta R)^2}{h^4} \right] \tag{3}$$

where $F$ is the force, $R$ is the radius of indenter, and $\delta$ is the indentation depth.

## Fluorescence imaging of giant unilamellar vesicles (GUVs)

GUVs with composition of DOPC: DOPS: Liss rhod PE(18:1) = 8: 2: 0.005 (Avanti Polar Lipids, Inc.) were produced via electroformation using the Vesicle Prep Pro (Nanion Technologies)[64]. In brief, 10 µl 2 mg/ml lipid mixtures in chloroform were deposited within an 18 mm O-ring chamber mounted on an indium tin oxide (ITO) electrode and placed in vacuum chamber for 2 hours. Subsequently, the O-ring chamber was filled with about 250 µl 300 mM sucrose solution (shielded from light with aluminum foil), and the Vesicle Prep Pro machine was operated using a standardized protocol (37 °C, 8 Hz, 1.1 V, 45 min Rise, 60 min Main, 45 min Fall). The resultant GUVs were stored in small glass bottle at 4 °C.

Before the addition of RCD-1 proteins, 90 µl mixed liquid (containing 80 µl GUVs with a composition of DOPC: DOPS: Liss rhod PE(18:1) = 8: 2: 0.005 in sucrose, 10 µl of 10 µM FITC-dextran (Sigma-Aldrich)) was dispensed on the glass-bottom dish. 10 µl 5 µM RCD-1 (RCD-1-1: RCD-1-2 = 1: 1) was then introduced to the mixed solution. The GUVs were imaged with a confocal fluorescence microscope (FV3000, Olympus) and processed with Fiji (ImageJ).

## Micropipette aspiration measurement of the bending modulus $k_c$

Using a glass micropipette controlled by a micromanipulator (Inject-Man 4, Eppendorf), we followed the previously reported methods for measuring the bending modulus of GUVs[31–33]. Simply, a single GUV was firstly held at the micropipette tip for ~2 minutes with slight suction pressure (~1 Pa in our experiments) to eliminate excess hidden membrane area and reduce thermal undulation effects. Subsequently, the pressure was gradually increased to ~300 Pa in ~5 Pa increments. To observe larger deformations, suction was sometimes further increased in 50 Pa increments, up to 1000 Pa.

The isotropic membrane tension ($\tau$) induced upon aspiration is related to the applied suction pressure ($\Delta P$) by:

$$\tau = \frac{\Delta P \cdot R_P}{2\left(1 - \frac{R_P}{R_V}\right)} \tag{4}$$

where $R_P$ and $R_V$ are the pipette and vesicle radii, respectively.

The aspiration length ($\Delta L$) increases with $\Delta P$ and is related to the observed apparent area strain ($\alpha_{app}$) as follows:

$$\alpha_{app} = \frac{\Delta A}{A_0} = \frac{2\pi R_P \Delta L}{A_0}\left(1 - \frac{R_P}{R_V}\right) = \frac{\Delta L\left[(R_P/R_V)^2 - (R_P/R_V)^3\right]}{2R_P} \tag{5}$$

where $A_0$ is the initial membrane area at low tension, and $A$ is the area after pressurization[31,32,37].

The bending modulus $k_c$ can be determined by following equation:

$$ln\frac{\tau}{\tau_0} = \left(\frac{8\pi k_c}{k_B T}\right)\alpha_{app} \tag{6}$$

where $k_B$ is the Boltzmann constant, $T$ is the absolute temperature, and $\tau_0$ is the reference tension[31]. The bending modulus $k_c$ is determined by plotting $ln(\tau)$ vs $\alpha_{app}$ at bending regime and taking the product of the slope with $k_B T/8\pi$[31–33]. Fig. S17 illustrates the measurement and calculation process for a GUV with composition of DOPC: DOPS = 8: 2.

## Reporting summary

Further information on research design is available in the Nature Portfolio Reporting Summary linked to this article.

## Data availability

The data that support this study are available from the corresponding authors upon request. The values from all bar graphs displayed in this manuscript as well as uncropped scans of all AFM images are supplied in the Source Data file. The cryo-EM structure of RCD-1 pore comprised by 11 dimers can be accessed by PDB under accession number 8JYZ. Source data are provided with this paper.

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

## Acknowledgements

We sincerely appreciate Prof. Feng Shao for the fruitful discussion and detailed suggestion of this work. This work was supported by the National Natural Science Foundation of China (NSFC 32371525, T2221001, 92353304, T2350011); the Strategic Priority Research Program of the Chinese Academy of Sciences (XDB37020105).

## Author contributions

F.J. designed and conceptualized the study. K.R. and F.J. performed the AFM and TEM experiments and analysed the data. K.R. and R.X. performed the micropipette aspiration experiments. X.G. performed the confocal fluorescence microscopy experiments. Y.L. expressed and purified the RCD-1 protein. X.L., Q.K., Q.F., F.Y. and J.D. analyzed and interpreted the results. F.J., J.D.F. and K.R. wrote the manuscript with input from all co-authors.

## Competing interests

The authors declare no competing interests.
