## [Transparent Peer Review file · Nature Communications]

Mechanisms of RCD-1 pore formation and membrane bending

Corresponding Author: Dr Fang Jiao

Version 0:

Reviewer comments:

Reviewer #1

(Remarks to the Author)

REVIEW: Mechanisms of RCD-1 pore formation and membrane bending

The authors present a well-rounded study of the assembly of RCD-1 protein using a variety of nanocharacterization techniques, but mainly focussed on atomic force microscopy. The work is thorough and very well presented. The manuscript provides a number of very interesting qualitative observations regarding the interaction of RCD-1 protein with membranes. I believe that these observations will be very interesting to people working in this field and warrants publication. Sometimes I feel that the interpretation of the data (especially the quantification of the bending rigidity cut-off of 18.3 k_bT), goes a bit beyond what the data warrants. The qualitative observations themselves are interesting and the data is solid, so I don't think that there is a need to go out on a limb with the interpretations. I therefore suggest the authors tone down their interpretations and let the readers make up their mind based on the provided data.

Detailed comments/questions:

- Why do the authors use ecoli total lipid if the origin of RCD-1 is of fungal origin?
- If the authors do not see a difference in height in the pore and pre-pore state, are they certain that these are two separate states? I normally associate pre-pore states with systems where the protein ring first assembles and then inserts in the membrane to form the pore. In fact, do the authors see real "pores" in their system? Figure 1b seems to show rings, but the center of the ring has the same height as the lipid membrane. So is there really a pore?
- Figure 1g: the authors say that this was done with high-temporal resolution AFM, but no temporal information is given. While I agree that it is tempting to draw the colored monomers in this figure, I find such manual interpretations quite risky. Especially some of the lines drawn in figS4 are not convincing. How many lines per pixel were these images taken? What was the processing that was done to the raw data? I would need to see multiple such images (two consecutive images of the same ring as well as images from multiple rings), ideally also scanned from different directions to trust this interpretation. If the authors can provide multiple such images great, if not I would simply suggest to remove this panel as it isn't necessary for the main message of the paper.
- The observation that there are two types of pores, mobile and stationary pores are very interesting. However, I do not understand how the authors do not see a difference in the height of the two if their interpretation is that one is inserted in the membrane and the other one is not? What part of the proteins inserts into the membrane in the stationary pore. Where are those parts in the mobile pore?
- I particularly like that the authors found assembly of stationary pores in Figure 2c. However, I do not see how they determine from when on they assume there is a hole formed in the membrane (sketches in figure 2c). While in Figure 2c-33min there is some hint of a darker region inside the ring, it isn't obvious to me in figure 2c-47min through 73 minutes. Can the authors provide a more thorough analysis of the height differences "inside" the pore and the background? Is the amount the inside is darker really commensurate with a hole in the bilayer? Can the author provide cross-sections and a z-color scale for this figure?
- The authors report they observe "bending" of the membrane as long as there are negatively charged lipid components in the mixture. This is also an interesting finding. However, I am not sure that the word "bending" is the appropriate word to use

here, since bending would imply that there is a change in curvature of the membrane. In the presented data, it seems that affected membranes are still flat, but are simply higher. Perhaps because some of the RCD-1 has slipped under the membrane (pure speculation)? In any case I don't know if there is enough evidence for a change in membrane strain and curvature to call this bending. However, I am not a membrane biophysicist, so I am not extremely familiar with the details of that field. Figure 3B shows the more conclusive evidence of membrane bending. I wonder however if there is such clear membrane bending in the GUVs, why are the "bent" areas in the AFM images all of the same height? Can the authors calculate an estimate for the bending moment the RCD-1 applies on the GUV?

- The authors "...conclude membrane bending is induced by the interaction with RCD-1 dimers." This statement is likely true, but is also very vague. Obviously there is no bending without the RCD-1, and there is bending with the RCD-1, so likely the bending is due to some interaction with the RCD-1. However, if the RCD-1 is in dimeric form or not can in my opinion not be conclusively stated. I suggest the authors either remove this claim or clearly mark it as speculation (it is OK to speculate).
- I appreciate that the authors try to link their observation of membrane bending to membrane stiffness. However, I have concerns about the measurements they do with AFM. The authors state: "To quantify our hypothesis that membrane rigidity determines the degree of membrane bending, we measured the bending moduli of some of the SLBs used in this work with AFM force spectroscopy" This is in my opinion not the ideal method. AFM force spectroscopy can only give a compression modulus. I wonder why the authors have not used more standard macroscopic techniques such as membrane aspiration through a capillary? The measured values have quite a spread. This is somewhat problematic since the authors later claim that there is a sharp cut-off between when there is membrane bending (bending modulus $<18.3k_b T$) and when no (bending modulus $>18.3k_b T$). I don't think this can be stated with such accuracy.

Reviewer #2

(Remarks to the Author)

Ren et al. report exciting new findings regarding the behavior of the cell death-inducing fungal gasdermin RCD-1. The authors have uncovered with the use of atomic force microscopy (AFM) that the fungal gasdermin forms heteromeric pores (composed of its two allelic variants RCD-1-1 and RCD-1-2) in vitro and describe the dynamics of the pore-formation and its dependence on the plasma membrane composition and rigidity.

I did find the manuscript overall well written and clear to understand. The reported results are solid, illuminating and contribute greatly to our understanding of this unusual (by its regulation) gasdermin protein.

I have a few additional comments below.

Minor Comments

Line 49: Maybe the word 'anomalous' could be changed as it carries a connotation of normativity, which seems inappropriate considering that both forms of activation are natural. Also, we don't really know if there are other GSDMs activated in similar fashion to RCD-1. It could be possible this sub-family is larger than expected initially.

Line 80: Have the authors tried to repeat the procedure/models with AlphaFold3, the recent release specialized to model protein complexes? Might be informative to see if the performances of the AI-based models have increased.

Lines 111-112: Could benefit from an added reference at the end of that phrase for the pre-pore states for GSDMD and GSDMA3.

Line 140: The 'suggesting' should be more grammatically correct by – "which suggests that".

Line 168: Real-time imaging of...

Line 299: '.. in addition to membrane acidity... '

Line 313: I would be more careful using 'pyroptosis in fungal cells' as for many from the mammalian field that could appear too permissive use of the term pyroptosis, depending on how exactly it is defined and what is prioritized in the definition. I would consider 'pyroptotic-like cell death' to be a more precise term to describe the RCD-1 and fungal gasdermins' cell death reaction.

Lines 340-344: Most importantly, RCD-1 also controls a rapid lytic cell death in *N. crassa*, which was the main reason and phenotype used to identify the gene.

Line 344: I think in the cited paper, there are also evidence for in vivo oligomers, although not really looking like an ordered pores.

I don't really have any major comments for the authors to address.

I will only add that it is slightly regrettable to not have an experiment with real fungal plasma membranes. That could have been informative as it would represent the authentic lipid bilayer RCD-1 is encountering. Especially considering that the authors haven't used phosphatidylinositols in the reconstitution of the artificial fungal plasma membrane. Not sure if this choice has been made based on technical reasons (impossibility to fix or extract pure enough fungal plasma membranes) or with other rationales in mind.

Version 1:

Reviewer comments:

Reviewer #1

(Remarks to the Author)

The authors have adequately addressed my previous comments.

Reviewer #2

(Remarks to the Author)

Nothing else to add to the review here, except that I consider the authors have addressed my mostly minor concerns. Thus I recommend 'acceptance' of the current manuscript.

In black the original reviewer comments.

In blue, our feedback to comments.

In yellow highlight, changes in main text made following reviewer's request.

Point-by-Point Response to Reviewer #1

Reviewer #1: The authors present a well-rounded study of the assembly of RCD-1 protein using a variety of nanocharacterization techniques, but mainly focused on atomic force microscopy. The work is thorough and very well presented.

Authors: We greatly appreciate the reviewer's insightful and constructive feedback, which has significantly contributed to improving our manuscript. Below, we provide detailed responses to each of the reviewer's comments.

Reviewer #1: The manuscript provides a number of very interesting qualitative observations regarding the interaction of RCD-1 protein with membranes. I believe that these observations will be very interesting to people working in this field and warrants publication. Sometimes I feel that the interpretation of the data (especially the quantification of the bending rigidity cut-off of 18.3 K_BT), goes a bit beyond what the data warrants. The qualitative observations themselves are interesting and the data is solid, so I don't think that there is a need to go out on a limb with the interpretations. I therefore suggest the authors tone down their interpretations and let the readers make up their mind based on the provided data.

Authors: We appreciate the reviewer's rigorous suggestions. Following the reviewer's advice, we have removed the descriptions of the minimum bending modulus values that lead to membrane bending. Instead, we now only provide a qualitative description of how the physical properties of the membrane influence membrane bending, identified with two characterization techniques: AFM and micropipette aspiration techniques.

Reviewer #1: Why do the authors use *E. coli* total lipid if the origin of RCD-1 is of fungal origin?

Authors: In the initial stage of this work, our goal was to investigate whether RCD-1 could form pores in negatively charged lipid membranes. This was based on previous

findings, which suggest that gasdermins (GSDMs) typically form pores in conjunction with negatively charged lipids¹⁻⁵, and the work by Asen Daskalov et al., which indicated that RCD-1 can interact with negatively charged phospholipids⁶.

Furthermore, lipids extracted from *E. coli* have been widely used as a model lipid component in studies involving various negatively charged phospholipids in GSDMs or other pore-forming proteins^{4,7-10}. Therefore, *E. coli* total lipid was selected in our initial experiments to investigate the pore-forming capabilities of RCD-1 in lipid membrane. After confirming that RCD-1 forms pores in *E. coli* total lipids, we varied the lipid components in subsequent sections.

In the revised manuscript, we have also included experiments using yeast extract polar lipids as a model for fungal lipid membranes. We found that RCD-1 forms pores and induces membrane bending in the yeast extract polar lipid membrane (Fig. S18), further supporting our conclusions.

Figure S18. AFM topographies of RCD-1 incubated with yeast extract polar membranes. AFM images show the incubation of (a) 100 nM RCD-1 and (b) 250 nM RCD-1 on yeast extract polar membranes, with bent membrane areas covering 36.5% and 67.8% in (a) and (b), respectively (a-b, experimental repeats, $n = 3$).

Reviewer #1: If the authors do not see a difference in height in the pore and pre-pore state, are they certain that these are two separate states? I normally associate pre-pore states with systems where the protein ring first assembles and then inserts in the membrane to form the pore. In fact, do the authors see real “pores” in their system? Figure 1b seems to show rings, but the center of the ring has the same height as the lipid membrane. So is there really a pore?

Authors: We appreciate the reviewer’s questions regarding the existence of RCD-1 pre-pore and pore state. We have updated the Z scale of height profiles in Figure 1b to better illustrate the center heights of RCD-1 pores are lower than the surrounding lipid

membrane, confirming their classification as genuine “pores”.

Based on the resolved structures^{3,9,11-13}, gasdermins (GSDMs) in both prepore and pore states exhibit nearly identical protrusion heights above the membrane, which makes them difficult to distinguish solely by height profiles (Fig. R1). We defined the RCD-1 oligomer states following the classifications established in reports on GSDMD and GSDMA3^{4,9}. Similar to RCD-1, no height collapses of the prepore and pore structures of either GSDMD or GSDMA3 were observed. The pore-forming states of GSDMs can be classified into three categories (as detailed in the Peer Review File of the report⁹): 1) Membrane-attached oligomers: These are weakly attached to the membrane surface, mobile, and don't form transmembrane pores, residing in a pre-pore state. 2) Membrane-inserted oligomers: These are inserted into the membrane, but the inner membrane is not yet released, also residing in the plugged pre-pore state. 3) Membrane-inserted oligomers forming open (lytic) transmembrane pores: These oligomers are in the pore state.

Thus, we add the descriptions: ‘Oligomers in the pore and pre-pore states were indistinguishable by height and mobility (Fig. 1b, 1c). Following previous reports on GSDMs^{4,5,22}, we categorize RCD-1 prepores as including both membrane-attached mobile oligomers, and membrane-inserted oligomers in the plugged pre-pore state, in which the inner membrane has not yet been released. RCD-1 pores consist of membrane-inserted oligomers that form open transmembrane pores with inner membrane released.’ (Line 109-113).

Figure Redacted

Figure R1. Cryo-EM structures of GSDMA3, GSDMD, and RCD-1^{3,11,14}. (a) (i) Superposition of the auto-inhibited form and the pore form of GSDMA3-NT. Structural transitions that accompany the formation of the two β -hairpins in the pore conformation³. (ii) Cryo-EM map (grey) superimposed onto the atomic model of the 27-fold symmetric GSDMA3 prepore (left) and pore (right)³. (b) (i) Superposition of the auto-inhibited form and the pore form of GSDMD¹¹. (ii) Ribbon diagram and dimensions of the 33-subunit GSDMD prepore (left) and pore (right) structures superimposed with its cryo-EM density¹¹. (c) (i) Comparison of inactive RCD-1-1 and RCD-1-2 structures with their pore subunit structures. Conformation-changed elements are labeled and highlighted by different colors¹⁴. (ii) Ribbon scheme and dimensions of the 22-subunit RCD-1 pore structure fitted into the cryo-EM density¹⁴.

Additionally, due to the small inner diameter of the RCD-1 ring (~13 nm) (Figure R1c) – the smallest known GSDM pore¹⁴ – and the tip size effect (Figure R2), the AFM

probe has difficulty fully contacting the bottom of the ring. This results in the inner center heights of the RCD-1 prepore and pore appear higher than their actual heights. Nevertheless, we can determine that the center hole of RCD-1 pore is approximately 1-2 nm lower than that of the surrounding lipid membrane.

Figure R2. Schematic illustration of AFM scanning of RCD-1 structures. The AFM tip is unable to touch the bottom of the ring hole due to the tip size effect, resulting in the center hole being only 1-2 nm lower than that of the surrounding lipid membrane.

Reviewer #1: Figure 1g: the authors say that this was done with high-temporal resolution AFM, but no temporal information is given. While I agree that it is tempting to draw the colored monomers in this figure, I find such manual interpretations quite risky. Especially some of the lines drawn in figS4 are not convincing. How many lines per pixel were these images taken? What was the processing that was done the raw data? I would need to see multiple such images (two consecutive images of the same ring as well as images from multiple rings), ideally also scanned from different directions to trust this interpretation. If the authors can provide multiple such images great, if not I would simply suggest to remove this panel as it isn't necessary for the main message of the paper.

Authors: We thank the reviewer for his/her insightful questions and suggestions. In our revised manuscript, we have replaced the previous AFM image with a new, higher-spatial resolution image in Fig.1g, which clearly exhibits many resolvable RCD-1 monomer structures. High spatial resolution images were collected at 512 x 512 pixels and a scan speed of 2 Hz over a 150 nm range. Linear flattening and 3 x 3 median filtering were applied to reduce noise. (Line 495-497).

In the revised manuscript, we added the description of 'Additionally, during the submission of our paper, the RCD-1 pore structure was resolved (Fig. 1g, upper right, Fig. S5)²⁸, showing that the RCD-1 pore is composed of 11 RCD-1-1/RCD-1-2 heterodimers, which falls into the range we observe for immobile rings (10 ± 2 dimers). Our high spatial-resolution topography also detected rings consisting of 5 dimers (Fig. 1g, Fig. S5d). Although we could not confirm whether these structures represent a pre-

pore or pore state, they are consistent with the AlphaFold3-predicted RCD-1 pore structures composed of 5 RCD-1-1/RCD-1-2 heterodimers (Fig. 1g, bottom right, Fig. S5). This suggests that RCD-1 may form very small pores comprising only 5 heterodimers.’ (Line 151-158).

Additionally, higher spatial-resolution images were consecutively collected in opposite directions, allowing for the resolution of monomers in both sets of images (Fig. S5).

Figure S5. High-resolution AFM images of 1 μ M RCD-1 oligomers on SLB made from 6:4 DOPE: CL (18: 1). Images were consecutively collected from opposite scan directions: (a) left to right and (b) right to left. (a-b, experimental repeats, $n = 3$). (c) Left, oligomers containing 11 dimers are highlighted within a green square. Middle, the height-section profile measured along the red line indicated in the topography. Right, recently resolved cryo-EM structure of the RCD-1 pore, composed of 11 RCD-1-1/RCD-1-2 heterodimers. (d) Left, oligomers containing 5 dimers are highlighted within an orange square. Middle, the height-section profile measured along the red line indicated in the topography. Right, AlphaFold3-

predicted RCD-1 pore structure formed by 5 RCD-1-1/RCD-1-2 heterodimers. RCD-1-1 is colored blue and RCD-1-2 is colored red.

Reviewer #1: The observation that there are two types of pores, mobile and stationary pores are very interesting. However, I do not understand how the authors do not see a difference in the height of the two if their interpretation is that one is inserted in the membrane and the other one is not? What part of the proteins inserts into the membrane in the stationary pore. Where are those parts in the mobile pore?

Authors: The reviewer misunderstood our description of RCD-1 rings as pores. We observed two types of RCD-1 rings: mobile rings and stationary rings. Based on the definitions of prepore and pore, which we updated in the revised manuscript (line 109-113), pre-pore RCD-1 includes both mobile rings and some stationary rings where inner lipid has not yet released, while all pores are stationary.

As shown in Fig. R1a, b, GSDMs in both prepore and pore states typically exhibit nearly identical protrusion heights above the membrane^{3,11}. Additionally, during the submission of our paper, the RCD-1 pore structure was resolved (Fig. R1c)¹⁴, both RCD-1-1 and RCD-1-2 exhibit conformational changes similar to those of GSDMD and GSDMA3 during the transition from prepore to pore. Thus, we infer the mobile rings are prepores with the transmembrane domain remaining as α helices, while stationary pores have transmembrane domains insert into the membrane as β sheets, with the globular domain outside the membrane remaining largely unchanged.

Reviewer #1: I particularly like that the authors found assembly of stationary pores in Figure 2c. However, I do not see how they determine from when on they assume there is a hole formed in the membrane (sketches in figure 2c). While in Figure 2c-33min there is some hint of a darker region inside the ring, it isn't obvious to me in figure 2c-47min through 73 minutes. Can the authors provide a more thorough analysis of the height differences "inside" the pore and the background? Is the amount the inside is darker really commensurate with a hole in the bilayer? Can the author provide cross-sections and a z-color scale for this figure?

Authors: We thank the reviewer for his/her thoughtful considerations and suggestions. In the revised manuscript, we have included the height section profiles of the RCD-1

oligomers and a Z-color scale for Figure 2c. As shown in Figure 2c, start from 33 minutes, we can determine that the center hole of RCD-1 pore is approximately 1-2 nm lower than that of the surrounding lipid membrane. The reason it is not deeper throughout the membrane is due to the tip size effect (Figure R2).

Fig. 2 Time-lapse topographs showing RCD-1 oligomerization and pore formation. (a) Time lapse topographies showing the high mobility of RCD-1 oligomers during oligomerization, with data collected by incubation of 125 nM RCD-1-1 and 125 nM RCD-1-2 on *E. coli* total SLB (experimental repeats, $n = 3$). The cartoons underneath to indicate the interpretation of growing oligomers. The yellow arrows indicate the same oligomers, the green arrow represents the scanning direction of AFM imaging. (b) The mobile rings transform into pores, with data collected by incubation of 500 nM RCD-1-1 and 500 nM RCD-1-2 on *E. coli* total SLB. The yellow arrowheads indicate the same oligomers (experimental repeats, $n = 3$). (c) Time-lapse topographs showing immobile RCD-1 oligomerization and pore formation, with

data collected by incubation of 125 nM RCD-1-1 and 125 nM RCD-1-2 on *E. coli* total SLB (experimental repeats, $n = 3$). The bottom panels show the height profile along the green dashed line. The yellow arrows indicate the same oligomers. The cartoons indicate the interpretation of growing pores.

Reviewer #1: The authors report they observe “bending” of the membrane as long as there are negatively charged lipid components in the mixture. This is also an interesting finding. However, I am not sure that the word “bending” is the appropriate word to use here, since bending would imply that there is a change in curvature of the membrane. In the presented data, it seems that affected membranes are still flat, but are simply higher. Perhaps because some of the RCD-1 has slipped under the membrane (pure speculation)? In any case I don’t know if there is enough evidence for a change in membrane strain and curvature to call this bending. However, I am not a membrane biophysicist, so I am not extremely familiar with the details of that field. Figure 3B shows the more conclusive evidence of membrane bending. I wonder however if there is such clear membrane bending in the GUVs, why are the “bent” areas in the AFM images all of the same height? Can the authors calculate an estimate for the bending moment the RCD-1 applies on the GUV?

Authors: The reviewer’s skepticism regarding whether the membrane is bending is reasonable. While we cannot exclude the possibility that some RCD-1 molecules may diffuse and slip under the membrane through defects during bending, it is unlikely to be the primary cause of the bending membrane.

There are several reasons for this:

1. As shown in Fig. 3b, membrane bending also occurred in the GUVs.
2. A two-step membrane bending was observed upon introducing a low concentration of RCD-1. As depicted in Figure 5, expansions of the bent area were observed with a height increase ~ 2 nm, followed by a subsequent height increase to a final height of 4 to 5 nm, accompanied by a reduction in the bent area size. If membrane bending were mediated by RCD-1 slipping underneath, a single-step bending event would likely be observed rather than the two-step bending behavior seen here.
3. As shown in Fig. S7, smaller bent areas adopt a sphere-like structure rather than a flat top when scanning under forces lower than 40 pN. Whether a sphere-like or flat defect is observed depends to some extent on the magnitude of the imaging force; when

using larger forces, the defects appear to be flatter (Fig. S7). To minimize parachuting artifacts that can produce long tails, we typically employ imaging forces ranging from 40 to 80 pN. (Line 222-226).

Together, we conclude that RCD-1 induces membrane bending by altering the curvature of the membrane.

Figure S7. Bent membrane imaged with various forces on an SLB made from DOPC: DOPS = 8: 2. (a) AFM images captured with scanning forces of 30, 40, 50, 80, 100 pN (experimental repeats, $n = 3$). (b) Height profiles along the dashed lines in (a).

To estimate the bending moment exerted by RCD-1 on the GUV, we employed micropipette aspiration to measure the bending modulus of the GUVs (Fig. S17). The curvature of a surface can be characterized by the average curvature ($1/R_1 + 1/R_2$), where R_1 and R_2 are the principal radii of curvature measured along two perpendicular directions³⁰. For a spherical surface, $R_1 = R_2$. The bending moment M of the curved membrane can be calculated using the formula:

$$M = k_c (1/R_1 + 1/R_2)$$

where k_c is the bending modulus. The method for calculating the bending modulus of GUVs follows previous studies³¹⁻³³. The measurements yielded a bending modulus of 18.1 ± 1.2 kBT for the GUVs formed by DOPC: DOPS = 8: 2, the same components used in Fig. 3b. The original radius of the GUV R_G is approximately $46 \mu\text{m}$, while the radius of the membrane bending region induced by RCD-1 R_B is about $15 \mu\text{m}$ (Fig. 3b). The change in curvature during this process is $(1/R_B - 1/R_G)$. Therefore, the bending moment exerted by RCD-1 on the GUV M_{RCD-1} can be calculated by $M_{RCD-1} = k_c \times (1/$

$R_B - 1/R_G$)¹⁵, resulting in a value of approximately $0.8 \text{ k}_B T/\mu\text{m}$. (Line 246-260).

Figure S17. Micropipette aspiration measurements of the bending modulus of GUVs composed of DOPC: DOPS = 8: 2. (a) Bright field images showing changes in GUV morphology with increasing suction pressure (experimental repeats, $n = 10$). (b) Tension-strain measurements for the GUV. (c) Linear fit of the low-tension regime in panel (b), where the slope (dashed lines) yielding the elastic bending moduli k_c ($\times 8\pi/k_B T$).

Reviewer #1: The authors “...conclude membrane bending is induced by the interaction with RCD-1 dimers.” This statement is likely true, but is also very vague. Obviously there is no bending without the RCD-1, and there is bending with the RCD-1, so likely the bending is due to some interaction with the RCD-1. However, if the RCD-1 is in dimeric form or not can in my opinion not be conclusively stated. I suggest the authors either remove this claim or clearly mark it as speculation (it is OK to speculate).

Authors: We appreciate the reviewer’s rigorous analysis and suggestion. Following the reviewer’s suggestion, we have revised the manuscript with ‘Thus, we speculate that membrane bending is induced by the interaction with RCD-1 dimers.’ (Line 238-239).

Reviewer #1: I appreciate that the authors try to link their observation of

membrane bending to membrane stiffness. However, I have concerns about the measurements they do with AFM. The authors state: “To quantify our hypothesis that membrane rigidity determines the degree of membrane bending, we measured the bending moduli of some of the SLBs used in this work with AFM force spectroscopy” This is in my opinion not the ideal method. AFM force spectroscopy can only give a compression modulus. I wonder why the authors have not used more standard macroscopic techniques such as membrane aspiration through a capillary? The measured values have quite a spread. This is somewhat problematic since the authors later claim that there is a sharp cut-off between when there is membrane bending (bending modulus $< 18.3 \text{ K}_B\text{T}$) and when no (bending modulus $> 18.3 \text{ K}_B\text{T}$). I don't think this can be stated with such accuracy.

Authors: We appreciate the review's concern and suggestions. Following the reviewer's advice, we employed micropipette aspiration to measure the bending modulus of GUVs, and only provided a qualitative description of how the physical properties of the membrane influence membrane bending (by removing the exact threshold value of bending modulus). In the revised manuscript, we have included the bending modulus results obtained from micropipette aspiration measurements in Fig. 6h and Fig. 7g and updated the description as following:

‘We also measured the bending moduli of GUVs with identical lipid compositions using micropipette aspiration by applying suction pressure to them (Fig. 6h, Fig. S17, Materials and Methods)³¹⁻³³. The values obtained from AFM and micropipette aspiration are qualitatively similar, although the average bending moduli measured by AFM were generally lower than those from micropipette aspiration (Fig. 6h). One possible explanation is that the GUV measurements using micropipette aspiration were conducted in a solution of 80% 100 mM sucrose and 20% 100 mM glucose, whereas our AFM measurements were performed in a salt buffer (20 mM HEPES-NaOH, pH 7.4, 150 mM NaCl). Numerous studies have shown that the bending modulus of acidic lipid membranes decreases with increasing salt ion concentration³⁹⁻⁴⁰. Given that all lipid compositions we consider contain an acidic lipid, this effect may account for the difference in bending moduli measured by the two techniques. However, both AFM and micropipette aspiration measurements consistently demonstrate that membrane rigidity significantly impacts the degree of membrane bending.’ (line 344-356).

Point-by-Point Response to Reviewer #2

Reviewer #2: Ren *et al.* report exciting new findings regarding the behavior of the cell death-inducing fungal gasdermin RCD-1. The authors have uncovered with the use of atomic force microscopy (AFM) that the fungal gasdermin forms heteromeric pores (composed of its two allelic variants RCD-1-1 and RCD-1-2) in vitro and describe the dynamics of the pore-formation and its dependence on the plasma membrane composition and rigidity.

I did find the manuscript overall well written and clear to understand. The reported results are solid, illuminating and contribute greatly to our understanding of this unusual (by its regulation) gasdermin protein.

Authors: We thank the reviewer for his/her wholly supportive and positive assessment of our paper.

Reviewer #2: Line 49: Maybe the word 'anomalous' could be changed as it carries a connotation of normativity, which seems inappropriate considering that both forms of activation are natural. Also, we don't really know if there are other GSDMs activated in similar fashion to RCD-1. It could be possible this sub-family is larger than expected initially.

Authors: We thank the reviewer's suggestion. In the revised manuscript, we have revised the word 'anomalous' with 'unusual' (line 55).

Reviewer #2: Line 80: Have the authors tried to repeat the procedure/models with AlphaFold3, the recent release specialized to model protein complexes? Might be informative to see if the performances of the AI-based models have increased.

Authors: We thank the reviewer for the professional suggestions. We have repeated the procedure/models using AlphaFold3 and included the results in Figure S2.

As shown in Figure S1 and S2, both AlphaFold2 and AlphaFold3 predicted diverse interaction structures for the homodimer (RCD-1-1/RCD-1-1 or RCD-1-2/RCD-1-2). The AlphaFold3 revealed very low-ranking scores of the top 5 structures, suggesting that both RCD-1-1/RCD-1-1 and RCD-1-2/RCD-1-2 homodimers are unlikely to form

stable oligomers. In contrast, both AlphaFold2 and AlphaFold3 predicted that the heterodimers (RCD-1-1/RCD-1-2) exhibits parallel arrangements of the β sheets, resembling mammalian GSDM pore structures. Unlike the varied structures predicted by AlphaFold2, the top 5 heterodimer structures predicted by AlphaFold3 show very similar results, closely matching the recently resolved RCD-1 pore structures¹⁴.

Figure S1. Predicted RCD-1 dimer structures by AlphaFold2-multimer. Predicted structures of (a) the RCD-1-1 homodimer, (b) the RCD-1-2 homodimer, and (c) the RCD-1-1/RCD-1-2 heterodimer. The width of RCD-1-1/RCD-1-2 heterodimer is approximately 5 nm. The five structures with highest pLDDT score are presented in order. RCD-1-1 is colored blue and RCD-1-2 is colored red.

Figure S2. Predicted RCD-1 dimer structures by AlphaFold3. Predicted structures of (a) the RCD-1-1 homodimer, (b) the RCD-1-2 homodimer, and (c) the RCD-1-1/RCD-1-2 heterodimer. The width of RCD-1-1/RCD-1-2 heterodimer is approximately 5 nm. The five structures with highest ranking score are presented in order. RCD-1-1 is colored blue and RCD-1-2 is colored red.

Reviewer #2: Lines 111-112: Could benefit from an added reference at the end of that phrase for the pre-pore states for GSDMD and GSDMA3.

Authors: we have added the references at the end of phrase ‘much like the pre-pore state reported for GSDMD and GSDMA3^{4,5,22}’ (line 121-122).

Reviewer #2: Line 140: The ‘suggesting’ should be more grammatically correct by – “which suggests that”.

Authors: We have replaced the ‘suggesting’ with ‘which suggests that’ in the revised manuscript (Line 148).

Reviewer #2: Line 168: Real-time imaging of...

Authors: We have updated the ‘Real-time imaging of RCD-1 oligomerization and pore formation’ in the revised manuscript (Line 189).

Reviewer #2: Line 299: ‘.. in addition to membrane acidity... ‘

Authors: We have revised the ‘The results also suggest that the pore formation of RCD-1 is related to membrane thickness in addition to membrane acidity.’ (Line 343).

Reviewer #2: Line 313: I would be more careful using ‘pyroptosis in fungal cells’ as for many from the mammalian field that could appear too permissive use of the term pyroptosis, depending on how exactly it is defined and what is prioritized in the definition. I would consider ‘pyroptotic-like cell death’ to be a more precise term to describe the RCD-1 and fungal gasdermins’ cell death reaction.

Authors: We have replaced the ‘pyroptosis in fungal cells’ with ‘pyroptotic-like cell death’ in the revised manuscript (Line 373).

Reviewer #2: Lines 340-344: Most importantly, RCD-1 also controls a rapid lytic cell death in *N. crassa*, which was the main reason and phenotype used to identify the gene.

Authors: We added the ‘Most importantly, RCD-1 also controls a rapid lytic cell death

in *N. crassa*, which was the main reason and phenotype used to identify the gene^{17,18},
in the revised manuscript (line 409-410).

Reviewer #2: Line 344: I think in the cited paper, there are also evidence for in vivo oligomers, although not really looking like an ordered pores.

Authors: We agree with the reviewer and have removed ‘in vitro’ from the end of the sentence in the revised manuscript (Line 414).

Reviewer #2: I don’t really have any major comments for the authors to address.

Authors: We sincerely appreciate the reviewer’s positive assessment and valuable suggestions for the paper.

Reviewer #2: I will only add that it is slightly regrettable to not have an experiment with real fungal plasma membranes. That could have been informative as it would represent the authentic lipid bilayer RCD-1 is encountering. Especially considering that the authors haven’t used phosphatidylinositols in the reconstitution of the artificial fungal plasma membrane. Not sure if this choice has been made based on technical reasons (impossibility to fix or extract pure enough fungal plasma membranes) or with other rationales in mind.

Authors: We appreciate the reviewer’s concern. Due to technical limitations, we have not yet obtained the natural cell membrane of *Neurospora crassa*. To address this, we investigated RCD-1 with one type of nature fungal cell membrane: the yeast extract polar membrane and described as following in our revised manuscript:

‘Additionally, we investigated RCD-1 with a real fungal cell membrane: the yeast extract polar membrane. As shown in Fig. S18, membrane bending also occurred when RCD-1 was co-incubated with this membrane. AFM and micropipette aspiration measurements indicate that the yeast extract polar membrane has a bending modulus close to the values observed for the model fungal cell membrane and the model modified fungal mitochondrial inner membrane (Fig. 7g). This result aligns with our findings from artificial lipid membranes, reinforcing the robustness of our conclusions.’
(Line 382-388).

Figure S18. AFM topographies of RCD-1 incubated with yeast extract polar membranes. AFM images show the incubation of (a) 100 nM RCD-1 and (b) 250 nM RCD-1 on yeast extract polar membranes, with bent membrane areas covering 36.5% and 67.8% in (a) and (b), respectively (a-b, experimental repeats, $n = 3$).

References

- 1 Liu, X. *et al.* Inflammasome-activated gasdermin D causes pyroptosis by forming membrane pores. *Nature* **535**, 153–158, doi:10.1038/nature18629 (2016).
- 2 Ding, J. *et al.* Pore-forming activity and structural autoinhibition of the gasdermin family. *Nature* **535**, 111–116, doi:10.1038/nature18590 (2016).
- 3 Ruan, J., Xia, S., Liu, X., Lieberman, J. & Wu, H. Cryo-EM structure of the gasdermin A3 membrane pore. *Nature* **557**, 62–67, doi:10.1038/s41586-018-0058-6 (2018).
- 4 Mulvihill, E. *et al.* Mechanism of membrane pore formation by human gasdermin-D. *The EMBO Journal* **37**, e98321, doi:<https://doi.org/10.15252/embj.201798321> (2018).
- 5 Broz, P., Pelegrín, P. & Shao, F. The gasdermins, a protein family executing cell death and inflammation. *Nature Reviews Immunology* **20**, 143–157, doi:10.1038/s41577-019-0228-2 (2020).
- 6 Daskalov, A., Mitchell, P. S., Sandstrom, A., Vance, R. E. & Glass, N. L. Molecular characterization of a fungal gasdermin-like protein. *Proceedings of the National Academy of Sciences* **117**, 18600–18607, doi:10.1073/pnas.2004876117 (2020).
- 7 Sborgi, L. *et al.* GSDMD membrane pore formation constitutes the mechanism of pyroptotic cell death. *The EMBO Journal* **35**, 1766–1778, doi:<https://doi.org/10.15252/embj.201694696> (2016).
- 8 Jiao, F. *et al.* Perforin-2 clockwise hand-over-hand pre-pore to pore transition mechanism. *Nature Communications* **13**, 5039, doi:10.1038/s41467-022-32757-4 (2022).

- 9 Mari, S. A. *et al.* Gasdermin-A3 pore formation propagates along variable pathways. *Nature Communications* **13**, 2609, doi:10.1038/s41467-022-30232-8 (2022).
- 10 Johnson, A. G. *et al.* Structure and assembly of a bacterial gasdermin pore. *Nature* **628**, 657-663, doi:10.1038/s41586-024-07216-3 (2024).
- 11 Xia, S. *et al.* Gasdermin D pore structure reveals preferential release of mature interleukin-1. *Nature* **593**, 607-611, doi:10.1038/s41586-021-03478-3 (2021).
- 12 Wang, C. *et al.* Structural basis for GSDMB pore formation and its targeting by IpaH7.8. *Nature* **616**, 590-597, doi:10.1038/s41586-023-05832-z (2023).
- 13 Zhong, X. *et al.* Structural mechanisms for regulation of GSDMB pore-forming activity. *Nature* **616**, 598-605, doi:10.1038/s41586-023-05872-5 (2023).
- 14 Li, Y. *et al.* Cleavage-independent activation of ancient eukaryotic gasdermins and structural mechanisms. *Science* **384**, adm9190, doi:doi:10.1126/science.adm9190 (2024).
- 15 Sackmann, E. Membrane bending energy concept of vesicle- and cell-shapes and shape-transitions. *FEBS Letters* **346**, 3-16, doi:[https://doi.org/10.1016/0014-5793\(94\)00484-6](https://doi.org/10.1016/0014-5793(94)00484-6) (1994).